# B-Type Natriuretic Peptide (BNP) Revisited—Is BNP Still a Biomarker for Heart Failure in the Angiotensin Receptor/Neprilysin Inhibitor Era?

**DOI:** 10.3390/biology11071034

**Published:** 2022-07-09

**Authors:** Toshio Nishikimi, Yasuaki Nakagawa

**Affiliations:** 1Department of Cardiovascular Medicine, Kyoto University Graduate School of Medicine, 54 Kawahara-cho, Shogoin, Sakyo-ku, Kyoto 606-8507, Japan; ynakagaw@kuhp.kyoto-u.ac.jp; 2Department of Medicine, Wakakusa Tatsuma Rehabilitation Hospital, 1580 Ooaza Tatsuma, Daito City 574-0012, Japan

**Keywords:** B-type natriuretic peptide, proBNP, heart failure, neprilysin, angiotensin receptor/neprilysin inhibitor

## Abstract

**Simple Summary:**

Active BNP-32, less active proBNP-108, and inactive N-terminal proBNP-76 all circulate in the blood. The circulating protease neprilysin has lower substrate specificity for BNP than ANP, while proBNP and N-terminal proBNP are not degraded by neprilysin. Currently available BNP immunoassays react with both mature BNP and proBNP; therefore, measured plasma BNP is mature BNP + proBNP. Because ARNI administration increases mature BNP, measured plasma BNP initially increases with ARNI administration by the amount of the increase in mature BNP. Later, ARNI administration reduces myocardial wall stress, and the resultant reduction in BNP production more than offsets the increase of mature BNP due to inhibition of degradation by neprilysin, resulting in lower plasma BNP levels. In the ARNI era, BNP remains a useful biomarker for heart failure, though mild increases early during ARNI administration should be taken into consideration.

**Abstract:**

Myocardial wall stress, cytokines, hormones, and ischemia all stimulate B-type (or brain) natriuretic peptide (BNP) gene expression. Within the myocardium, ProBNP-108, a BNP precursor, undergoes glycosylation, after which a portion is cleaved by furin into mature BNP-32 and N-terminal proBNP-76, depending on the glycosylation status. As a result, active BNP, less active proBNP, and inactive N-terminal proBNP all circulate in the blood. There are three major pathways for BNP clearance: (1) cellular internalization via natriuretic peptide receptor (NPR)-A and NPR-C; (2) degradation by proteases in the blood, including neprilysin, dipeptidyl-peptidase-IV, insulin degrading enzyme, etc.; and (3) excretion in the urine. Because neprilysin has lower substrate specificity for BNP than atrial natriuretic peptide (ANP), the increase in plasma BNP after angiotensin receptor neprilysin inhibitor (ARNI) administration is much smaller than the increase in plasma ANP. Currently available BNP immunoassays react with both mature BNP and proBNP. Therefore, BNP measured with an immunoassay is mature BNP + proBNP. ARNI administration increases mature BNP but not proBNP, as the latter is not degraded by neprilysin. Consequently, measured plasma BNP initially increases with ARNI administration by the amount of the increase in mature BNP. Later, ARNI reduces myocardial wall stress, and the resultant reduction in BNP production more than offsets the increase in mature BNP mediated by inhibiting degradation by neprilysin, which lowers plasma BNP levels. These results suggest that even in the ARNI era, BNP can be used for diagnosis and assessment of the pathophysiology and prognosis of heart failure, though the mild increases early during ARNI administration should be taken into consideration.

## 1. Introduction

B-type (or brain) natriuretic peptide (BNP) is a cardiac-derived peptide [1], although it was first discovered in the porcine brain [2]. Plasma BNP levels in normal subjects are about 0–20 pg/mL and plasma BNP levels are increased in left ventricular hypertrophy [3], myocardial infarction [4,5], coronary artery disease [6], pulmonary hypertension [7], and heart failure [8,9], according to the severity of the disease. BNP is currently used as a biochemical marker for heart failure in clinical settings because it reflects the state of heart failure extremely well [8,9]. Accordingly, heart failure guidelines in Europe [10], North America [11], and around the world now recommend measuring BNP/NT-proBNP for the diagnosis of heart failure in patients with shortness of breath. Many studies have revealed that NT-proBNP is as useful as BNP for the diagnosis of heart failure [12,13], and this is because proBNP is cleaved to BNP and NT-proBNP, and the heart secretes BNP and NT-proBNP in an equimolar fashion. However, since metabolism of BNP in plasma is different from that in NT-proBNP (See Section 5), NT-proBNP is easily influenced by age [14]. For example, Januzzi et al. [15] previously reported that since NT-proBNP is greatly influenced by the age, an optimal strategy to identify acute heart failure using NT-proBNP was to use age-related cutoffs of 450, 900, and 1800 pg/mL for ages < 50, 50 to 75, and >75, respectively. On the other hand, in the diagnosis of acute heart failure using BNP, the cutoff point is not age dependent [16,17,18]. Furthermore, since NT-proBNP is easily influenced by renal function, it is important to note that even mild renal impairment may overestimate the true NT-proBNP value (See Section 5).

The PARADIGM-HF trial showed that among chronic heart failure patients with reduced systolic function, sacubitril/valsartan, an angiotensin receptor neprilysin inhibitor (ARNI), elicits significantly greater decreases than enalapril in incidences of cardiovascular death and initial hospitalization for heart failure (the study’s combined endpoint) as well as total mortality [19]. Based on these results, recent ESC and ACC/AHA guidelines recommend the use of ARNI as a first-line treatment for chronic heart failure [10,11]. Because one of the functions of neprilysin is to degrade BNP, it has been suggested that plasma BNP levels may increase when ARNI is used and that, as a result, BNP may no longer reflect the state of the heart failure [20]. The purpose of this review is to reconsider the significance of BNP as a marker for heart failure in the ARNI era based on the current knowledge of the biochemistry and molecular biology of BNP, the immunoassay methodology used for BNP, the physiological action of neprilysin toward BNP, and the results of recent clinical studies.

## 2. Structure of the BNP Gene and the Amino Acid Sequence of BNP

BNP was first isolated from extracts of porcine brain in 1988 [2]. Shortly after its discovery, the highest tissue levels of BNP were shown to be in the heart [21], where it acts as a cardiac hormone [1,8]. Since then, BNP peptides and cDNA clones from various species have been isolated and sequenced. The predominant circulating molecular form of BNP is BNP-26 in pigs, BNP-45 in rats, and BNP-32 in humans (Figure 1) [22,23]. The reason for the difference in these molecules becomes apparent upon examination of their NH_2_-terminal regions. The NH_2_-terminal region of human proBNP contains the consensus sequence R^73^-X^74^-X^75^-R^76^, which is cleaved by furin. In the rat NH_2_-terminal region of proBNP, one amino acid is deleted between R^61^ and R^63^ as compared with the corresponding human sequence (R^61^-X^62^-R^63^ vs. R^73^-X^74^-X^75^-R^76^). As a result, the rat sequence cannot function as a cleavage site for furin. Additionally, human E^59^ corresponds to rat R^47^, which contributes to the R^47^-X^48^-X^49^-R^50^ consensus sequence. Consequently, rat proBNP is cleaved by furin at R^50^-S^51^ to form BNP-45, while human proBNP is cleaved by furin at R^76^-S^77^ to form BNP-32 [24]. These differences mean the structure of BNP varies among species, whereas the structure of atrial natriuretic peptide (ANP) is comparatively well conserved. The mature active molecular form of human BNP is BNP-32 (Figure 1). The BNP gene is mapped on chromosome 1 in humans and contains three exons and two introns (Figure 2) [25]. Exon 1 encodes a 26 amino acid signal peptide and the first 15 amino acids of proBNP. Exon 2 encodes most of the proBNP sequence, and exon 3 encodes the terminal tyrosine and the 3′-untranslated region. BNP mRNA is spliced to produce the mature BNP mRNA, which crosses the nuclear membrane and is translated into 134 amino acid preproBNP in the endoplasmic reticulum. The 26 amino acid signal peptide is removed to produce proBNP-108 [26,27], which is transported to the Golgi apparatus, where it undergoes *O*-glycosylation at several sites [28]. Whereas atrial and ventricular tissue contain only precursor proANP, they contain the precursor, mature, and inactive forms of BNP: proBNP-108, BNP-32, and N-terminal (NT)-proBNP-76. Mature BNP-32 is predominant in atrial tissue (~60%), while proBNP-108 predominates in ventricular tissue (~60%) [29]. Processing of proBNP to BNP and NT-proBNP occurs within the trans-Golgi network [30] by furin, and it had been thought that mature BNP and NT-proBNP are then released into the blood via a constitutive pathway. However, recent studies have shown that uncleaved proBNP also circulates in the blood [31,32,33], is glycosylated [34,35], and increases as heart failure becomes more severe [36]. Those findings indicate that all proBNP is not cleaved and that glycosidic residues attached to proBNP at specific sites may affect the efficiency of proBNP processing [36,37]. The mechanism of proBNP processing is discussed in detail in Section 4.

## 3. Localization and mRNA Expression of BNP

Although the concentration of BNP and its mRNA in the human ventricle is much lower than in the atria, the total amount of BNP and its mRNA in the ventricle respectively account for 30% and 70% of the total in heart [8]. Clinical studies have also shown that plasma BNP concentrations are higher in the anterior interventricular vein and coronary sinus than in the aortic root, demonstrating that BNP is a cardiac hormone synthesized primarily in the ventricles and secreted from them [9]. Furthermore, no detectable levels of BNP are found in rat or human brain, suggesting that the tissue distribution of BNP varies among species [38]. ANP is stored in secretory granules as proANP. When the atrium is stimulated, proANP is cleaved into ANP and NT-proANP by the membrane-bound enzyme corin and released into the blood [24,39,40]. By contrast, within the myocardium proBNP is cleaved into BNP and NT-proBNP by the processing enzyme furin in the trans-Golgi network, after which both are constitutively released into the circulation (Figure 2) [24,30,41]. Interestingly, recent studies showed that proBNP is not entirely cleaved into BNP and NT-proBNP; instead, a significant portion of proBNP is released into the blood unchanged [29,31,32,33,42]. In congestive heart failure patients, myocardial BNP mRNA and plasma BNP and NT-proBNP levels are greatly increased as compared to ANP levels [8,9], indicating that BNP functions as an emergency defense against ventricular overload under pathological conditions. BNP mRNA is characterized by the presence of a conserved sequence consisting of a repeated AUUUA unit in the 3′-untranslated region (Figure 2), which is not observed in ANP mRNA. The presence of this sequence promotes degradation of BNP mRNA in a manner similar to that observed with lymphokine genes and oncogenes [43]. Thus, BNP gene expression is regulated differently from ANP gene expression and is thought to be dynamically altered by physiological and pathological conditions.

## 4. Transcriptional Regulation of BNP Gene

It is well known that the BNP gene expression is tissue specific and developmentally regulated. Many studies have demonstrated the importance of cis-acting elements responsible for controlling basal and inducible BNP gene transcription [44,45,46]. For example, in a study using transgenic mice, coupling a segment of the 5′-flanking region (FR) of human BNP (−1818 to +100 or −408 to +100) to the luciferase gene showed that the proximal region of the human BNP promoter is sufficient to mediate ventricle-specific expression (Figure 3) [47]. In addition, the luciferase activity of −1818hBNPluc was also larger in ventricular than atrial myocytes, and deletion analysis revealed that expression of a segment of the human BNP 5′-FR from −127 to −40 is heart-specific [48]. There are potential GATA-, M-CAT-, and AP-1/CRE-like elements in the proximal region of the human BNP promoter, which are conserved among humans, rats, and mice. These three elements regulate heart-specific gene expression and also mediate both basal and induced BNP gene expression [49]. GATA-4 regulates expression of many heart-specific genes, and mechanical stretch transiently increases GATA-4 DNA binding and transcript levels in cultured cardiac myocytes [50]. AP-1 is the binding site for the products of two proto-oncogenes, c-*fos* and c-*jun*. AP-1 binding activity is increased in an animal model of aortic coarctation-induced pressure overload in the heart [51]. A number of studies have revealed that other sites relatively distal to the human BNP 5′-FR, such as neuron-restrictive silencer element (NRSE) (−552), shear stress-responsive elements (SSRE) (−652, −641 and −161), thyroid hormone-responsive element (TRE) (−1000), and the nuclear factor of activated T-cell (NFAT) binding site (−927), are also involved in the inducible activation of the human BNP promoter [44,49,52]. SSRE, located in the proximal promotor at the −161-bp, −641-bp, and −652-bp positions, is thought to be partly responsible for the induction of human BNP by mechanical stress. Single or multiple site-directed mutations of SSRE elements or co-transfection with a constitutive suppressor of NF-kB activity caused a maximum 40% decrease in strain-activated human BNP promotor activity [53]. Among these elements, the transcriptional repressor element NRSE, located at −552 of the human BNP promoter represses basal BNP promoter activity and mediates cardiac hypertrophic signaling induced by fibronectin [54]. Neuron restrictive-silencer factor (NRSF) binds to NRSE, thereby repressing promoter activity. Moreover, heart-restricted inactivation of NRSF by over expression of a dominant-negative NRSF driven by the cardiac-specific α-MHC promoter increased mRNA expression of ANP and BNP in the ventricle and induced cardiac dysfunction and sudden death, which confirms the importance of NRSF in the molecular mechanism of failing heart [55]. Thus, under normal conditions, NRSF suppresses the fetal myocardial gene program, thereby inhibiting maladaptive cardiac remodeling. However, a recent study showed that under pathological conditions NRSF nuclear repressor function is attenuated by CaMKII-dependent phosphorylation of class II HDAC54, allowing reactivation of the fetal cellular gene program [56]. In addition, cardiac expression of Gαo, an inhibitory G protein encoded in humans by GNAO1, which is transcriptionally regulated by NRSF, was increased in the ventricles of several heart failure models [56].

## 5. Production, Processing, Secretion, and Metabolism of BNP

To more accurately interpret the meaning of plasma BNP concentrations, it is important to better understand BNP production, processing, secretion, and metabolism (Figure 4) [41]. When cardiomyocytes are subjected to a stimulus such as mechanical stress, pressure or volume overload, certain hormones (e.g., angiotensin II or endothelin), cytokines, or ischemia, BNP gene expression is transcriptionally upregulated through actions of the aforementioned transcription factors [44,45,46], and the resulting transcript is translated in the endoplasmic reticulum to produce preproBNP [26,27]. The signal peptide is then removed from preproBNP by signal peptidase, yielding proBNP, which is transported to the Golgi apparatus, where it is glycosylated to various degrees at seven sites in its N-terminal region: threonine (Thr)36, serine (Ser)37, Ser44, Thr48, Ser53, Thr58, and Thr71 [28]. The glycosylated proBNP is then transported to the trans-Golgi network, where it had been thought that the glycosylated proBNP is cleaved into BNP and NT-proBNP by the processing enzyme furin [24]. However, it is now known that a considerable amount of glycosylated proBNP is not cleaved but is instead secreted intact and circulates in the blood (Figure 2 and Figure 4) [29,34,35]. Moreover, recent studies by us and others have shown that it is *O*-linked glycosylation at Thr48 and Thr71 in the N-terminal portion of proBNP catalyzed, at least in part, by polypeptide N-acetylgalactosaminyltransferase 2 that induces resistance to processing by furin convertase [36,37]. As a result, BNP, NT-proBNP, and proBNP are all secreted into the blood and are subject to different metabolic pathways [57].

Circulating BNP binds to natriuretic peptide receptor-A (NPR-A), which is expressed in the kidney, vascular smooth muscle, adrenal gland, brain, heart, testis, eye, intestine, and olfactory mucosa [58]. This receptor is bound to an intracellular guanylate cyclase domain, so that upon ligand binding, cGMP is produced intracellularly, and signal transduction is evoked [59]. In renal tubule cells, for example, this elicits natriuretic and diuretic action, while in renal blood vessels it elicits dilation. The bound BNP is cleared through internalization of the receptor and intracellular metabolism. Natriuretic peptide receptor-C (NPR-C) represents 95% of all natriuretic peptide receptors and is expressed in the kidneys, adrenal glands, lungs, vascular wall, intestine, brain, and all chambers of the heart [60]. Because no guanylate cyclase domain is bound to NPR-C and no cGMP is produced when a ligand binds to the receptor, NPR-C is considered to function as a clearance receptor. On the other hand, several studies have shown that NPR-C is coupled to adenylyl cyclase inhibition or phospholipase C activation through the inhibitory guanine nucleotide regulatory protein (Gi) [61,62,63]. NT-proBNP does not bind to any receptors, so metabolism via these receptors does not occur, while ProBNP exhibits only weak receptor binding [34,36]. However, there are numerous proteases in the blood that degrade these and other peptides. It is known that dipeptidyl-peptidase-IV, insulin degrading enzyme, meprin A, neprilysin, and others are all involved in degrading BNP [64,65,66,67]. Among those, neprilysin reportedly degrades BNP but not NT-proBNP or proBNP [68].

Excretion from the kidney is also an important means of clearing molecules that are not enzymatically degraded within the blood or a receptor-mediated pathway. BNP and NT-proBNP are readily filtered from the glomerulus and excreted in urine [69]. Because proBNP has a molecular weight of 12,000, it was thought to be excreted in urine unchanged. In reality, however, because proBNP is glycosylated, its molecular weight is actually 20,000 to 30,000, making its glomerular filtration difficult. When we used our newly developed immunoassay to measure plasma levels of mature BNP and proBNP [70] in samples collected from the aorta and renal vein in heart failure patients, the mature BNP concentration in renal venous plasma was much lower than in arterial plasma, while the change in the proBNP concentration between arterial and renal venous blood was smaller. This indicates there is less excretion of proBNP in the urine and/or less receptor-mediated internalization (unpublished observation). Thus, mature BNP is metabolized via various pathways, including internalization after receptor binding, degradation by enzymes in the blood, and excretion in the urine. Whereas clearing NT-proBNP depends nearly entirely on excretion in the urine [57]. Consequently, NT-proBNP is known to increase with even mildly impaired renal function, and NT-proBNP is increased to a much greater degree than BNP in renal failure [71]. Clearance of proBNP exhibits some of the characteristics of both BNP and NT-proBNP; that is, it exhibits weak binding to natriuretic receptors, no neprilysin substrate specificity, and little excretion via the kidney. The plasma half-life of BNP in humans is approximately 18 min, whereas that of ANP is approximately 3 min [72,73]. The difference in half-life between the two peptides may be due to lower binding affinity of BNP for NPR-C [74] and its higher resistance to neprilysin [75,76].

## 6. Substrate Specificity of Neprilysin for ANP and BNP

Neprilysin is a widely expressed protease that localizes to the plasma membrane in various cell types and is highly expressed in the brush borders of renal tubules [77]. Among neprilysin’s substrates are the natriuretic peptide family (ANP, BNP, and CNP) angiotensin II and endothelin, all of which are associated with the pathophysiology of heart failure [78]. Among the natriuretic peptide family, neprilysin has high substrate specificity for ANP. For instance, although ANP stimulates cGMP formation in porcine LLC-PK1 cells, if ANP is first incubated with brush border membranes, its ability to induce cGMP formation is remarkedly reduced. On the other hand, the same treatment does not affect BNP-induced cGMP formation [75]. ANP is thus degraded by neprilysin in the brush border membrane, but BNP is much less affected. Moreover, upon incubation of ANP, BNP, or CNP with neprilysin purified from human kidney, about 80% of ANP and CNP are degraded within 4 h, whereas only about 20% of BNP is degraded over the same time period [76]. These findings indicate that while neprilysin has high substrate specificity for ANP and CNP, its substrate specificity for BNP is low. The three-dimensional X-ray structure of neprilysin bound with a cyclic fragment of ANP, BNP, or CNP provides the basis for a model of the interaction between natriuretic peptides and neprilysin that explains the difference in substrate specificity [79]. According to this model, the natriuretic peptide slips into an interior cavity of neprilysin likely through a molecular sieve-like opening formed by the C-terminal catalytic thermolysin-like domain and the smaller N-terminal domain. Notably, the long N-terminus of BNP leads to spatial clashes, hindering its proper orientation with respect to the catalytic site. Missing the C-terminal extension (CNP) and having a short N-terminal region (CNP and ANP) favors binding within the cavity. Thus neprilysin-catalyzed degradation of natriuretic peptides is dependent on the length of the peptide’s N- and C-termini. For that reason, mouse ANP is rapidly degraded when incubated with neprilysin, but human BNP is not. Moreover, incubation of neprilysin with combined mouse ANP and human BNP attenuates degradation of the ANP, suggesting human BNP can act an endogenous inhibitor of neprilysin [79]. Consistent with that idea, it was observed that when plasma BNP rose above 916 pg/mL in patients with heart failure, neprilysin activity was markedly reduced. In addition, 95% of the patients in that study could be stratified into two groups: BNP < 916 pg/mL and neprilysin activity > 0.21 nmol/mL/min, and BNP > 916 pg/mL and neprilysin activity < 0.21 nmol/mL/min. Furthermore, adding synthetic BNP to plasma samples led to inhibition of neprilysin activity. Collectively, these results suggest that BNP acts as an endogenous neprilysin inhibitor in heart failure patients with markedly increased BNP levels [80].

A recent study in which sacubitril/valsartan was administered to heart failure patients to assess its effects on plasma ANP and urinary cGMP levels showed that, after 2 weeks of treatment, plasma ANP levels had increased approximately 1.5-fold, and there was also an increase in urinary cGMP levels that correlated well with plasma ANP but not BNP [81]. Echocardiographic examination revealed that the initial increase in ANP levels correlated with the magnitude of the reduction in left ventricular ejection fraction (LVEF) and left atrial volume. These results suggest that sacubitril/valsartan mainly increases plasma ANP, and its left ventricular reverse remodeling may be mediated in part via an increase in ANP/cGMP signaling [81].

In addition, it was also shown that 1 day after administration of a neprilysin inhibitor (SCH42495), neprilysin activity was reduced by 90% or more, plasma ANP concentrations were increased, and plasma BNP concentrations also tended to increase. Both natriuresis and diuresis were subsequently observed, while blood pressure decreased slightly. On day 4 of administration, plasma BNP concentrations were significantly lower than control, whereas plasma ANP tended to decrease but remained nearly the same as the control [82]. These results indicate that inhibiting neprilysin reduces myocardial wall stress through natriuresis, diuresis, and vasodilation, which explains the reduction in plasma BNP. Although BNP degradation was slightly weakened by neprilysin inhibition, the decrease in myocardial wall stress had a stronger effect reducing BNP, leading to a decrease in plasma BNP.

Nougué et al. [83] recently reported that ARNI treatment for 30 days produced an ∼4-fold increase in plasma ANP levels and a mild decrease in NT-proBNP levels without changing BNP levels. They also found a significant relationship between the BNP and NT-proBNP levels. This suggests the beneficial effects of ARNI are likely mediated by ANP and other substances that are substrates for neprilysin but are unlikely to be mediated by BNP [84].

In summary, sacubitril/valsartan combines the actions of both an angiotensin II receptor blocker and a neprilysin inhibitor, thereby reducing myocardial wall stress more effectively than either agent alone. When ARNI-induced reductions in myocardial wall stress are sufficient, BNP gene expression and production are reduced, leading to reduced plasma BNP levels despite the neprilysin inhibition.

## 7. Changes in BNP after ARNI Administration in a Clinical Study

In the PIONEER-HF trial, sacubitril/valsartan or enalapril was administered after hemodynamics had been stabilized in acute decompensated heart failure patients, and the effects of the two agents were compared based on NT-proBNP levels. Sacubitril/valsartan reduced NT-proBNP more than enalapril did (47% vs. 25% from baseline), whereas BNP was decreased by 33% with enalapril and by 29% with sacubitril/valsartan. This study thus provides an example in which BNP was reduced when administration of ARNI induced a sufficient reduction in myocardial wall stress [85].

In the PARADIGM-HF trial [19], the percentage change in BNP and NT-proBNP from the pretreatment levels showed that the median BNP level had increased by 19% after 8 to 10 weeks of treatment, while the median NT-proBNP level had decreased by 28%. Nevertheless, there was a strong positive correlation between BNP and NT-proBNP levels (r = 0.75). Good positive correlations were also observed between BNP and NT-proBNP before treatment, after 4–6 weeks of treatment, and after 9 months of treatment. BNP and NT-proBNP also had comparable prognostic performances, and the concentrations of BNP and NT-proBNP were strongly associated with the primary endpoint at all time points after ARNI treatment, independent of key demographic and clinical factors [86]. These results suggest that BNP and NT-proBNP provide similar information about heart failure status and risk stratification, even during ARNI treatment.

A recent study also showed that six months of ARNI administration significantly reduced plasma BNP levels from 576 pg/mL to 367 pg/mL in 70 heart failure patients with reduced ejection fraction [87]. The heart failure re-admission rate in the “descending group”, which showed a decrease of 50 pg/mL or more in BNP after 6 months of ARNI administration, was only about 21%, whereas the “rising group”, which showed an increase in BNP of 50 pg/mL or more after 6 months of ARNI administration, was about 50%. This indicates that if ARNI treatment sufficiently lowers myocardial wall stress in heart failure patients, the decrease in BNP may provide a good prognostic marker. Consistent with those studies, another recent study showed that ARNI treatment significantly reduced plasma BNP levels from 181 pg/mL to 70 pg/mL with corresponding improvements in exercise capacity and LVEF and a reduction in furosemide dose [88]. In a case report, after stabilization of hemodynamics in acute heart failure, ARNI treatment led to parallel reductions in BNP and NT-proBNP [89]. Furthermore, a recent study in which mass spectrometry was used to accurately measure active BNP-32 concentrations showed that BNP-32 accounts for only 20% to 25% of BNP levels measured by conventional clinical immunoassays and that BNP-32 levels are significantly decreased after 8 weeks of treatment with ARNI [90].

Collectively, these results suggest that decreasing myocardial wall stress with ARNI diminishes the stimulus for BNP production to an extent that more than offsets any increase in plasma BNP caused by inhibiting neprilysin-catalyzed degradation.

## 8. Immunoreactive BNP in the Plasma Means “proBNP + Mature BNP”

The method currently used to measure plasma BNP levels in a clinical setting is the immunoassay [91,92,93]. However, recent studies have shown that there are marked differences in the measured concentration of BNP obtained with different immunoassay kits [94,95]. Indeed, BNP concentrations measured in 40 plasma samples from heart failure patients differed considerably among the five available BNP immunoassays [96]. Consistent with that finding, another recent study showed that changes in plasma BNP measured in patients with chronic heart failure treated with ARNI differ considerably depending on the immunoassay used [97]. This means that when assessing changes in plasma BNP levels, it is important to always use the same BNP assay kit.

When interpreting plasma BNP levels measured with the currently available immunoassay kits, it is necessary to understand the true meaning of the BNP levels measured. Most of the immunoassays now being used worldwide entail sandwiching a BNP molecule between the two antibodies (Table 1) [91,92,93,94,95,96]. Importantly, BNP levels measured with these immunoassays are little affected by the N-terminal length of the BNP molecule. Consequently, when proBNP is present in the plasma [29,31,32,33], the immunoassay cross-reacts with it and measures it as BNP, and the extent of this cross-reactivity differs among these assays [98]. As a result, the BNP values measured with the currently available BNP immunoassay kits are, to varying degrees, actually “BNP + proBNP” [99].

When treating patients with heart failure, it is important to know the proBNP level and/or the proBNP/total BNP ratio because proBNP has less cGMP-producing activity than BNP, and an increase in proBNP may be associated with the worsening heart failure or development of heart failure [34,36]. However, there are few methods for measuring the proBNP/total BNP ratio. We previously measured proBNP/total BNP ratios using a combination of gel-filtration and a fluorescent immunoenzyme assay with BNP and found that the proBNP/total BNP ratio varies widely among heart failure patients [29]. However, the method used in that study requires a great deal of time and effort, and extraction of the peptide from plasma may cause underestimation of the proBNP levels due to its high adsorption on the extraction column [100], which reduces the recovered fraction of the peptide. To solve these problems, we developed a highly sensitive and accurate method to measure plasma proBNP and total BNP levels more quickly and easily. Our idea was to make a sandwich immunoassay using a capture antibody that recognizes the C-terminal region of both BNP and proBNP and two detection antibodies that respectively recognize the N-terminal region of proBNP and the ring structure of BNP with similar affinities (Figure 5) [70]. Using this approach, we were able to develop a sensitive immunochemiluminescent assay for proBNP and total BNP in plasma and calculate mature BNP precisely (mature BNP = total BNP − proBNP). Mature BNP includes BNP1-32 (BNP-32) as well as BNP3-32, BNP4-32, and BNP5-32 due to their antibody recognition sites. They are known to be present in blood and are thought to have biological activity [101,102]. Using these kits, we measured total plasma BNP and proBNP in healthy subjects and heart failure patients. We found that approximately 60–70% of the total BNP in venous plasma was proBNP and that 30–40% was mature BNP in both healthy subjects and heart failure patients [70]. In other words, the major molecular form of immunoreactive BNP in human venous blood is proBNP. The smaller proportion of mature BNP among the immunoreactive BNP forms has also been observed with mass spectrometry [90]. In addition, the proBNP/total BNP ratio increases with the severity of chronic heart failure [36]. Conversely, the mature BNP/total BNP ratio increases in patients with mild acute heart failure compensating by elevating mature BNP/cGMP signaling [103,104]. More recently, Kimura et al. [105] showed that in acute decompensated heart failure patients with reduced ejection fraction, the mature BNP/total BNP ratio on admission is associated with reverse remodeling capacity. In heart failure, *O-*glycosylation of specific amino acid residues in the N-terminal region of proBNP may be altered to regulate processing efficiency, and the amount of the active form may be fine-tuned according to the pathophysiological condition. Therefore, when evaluating plasma BNP levels before and after ARNI with a currently available BNP immunoassay, it is necessary to keep in mind that BNP means mature BNP + proBNP and interpret it accordingly.

## 9. The Change of “Mature BNP” during ARNI Treatment—Case Presentation

We now present three cases of heart failure in which the patients were treated with ARNI, and we were able to measure total BNP, proBNP and mature BNP before and after treatment. Case 1 was a 50-year-old male patient with a history of diabetes, dyslipidemia, and smoking habits. He had an acute myocardial infarction (antero-septal infarction) about 6 months prior and underwent percutaneous coronary intervention (PCI) on an anterior descending artery lesion. He was admitted to the hospital because of heart failure. After symptoms and signs improved by the treatment, rehabilitation was started. Echocardiography revealed his LVEF to be 35%. Because plasma BNP levels gradually increased, ARNI (50 mg bid) was administered, and plasma total BNP, mature BNP, and proBNP were measured before and 2 and 5 weeks after initiation of ARNI treatment. After 2 weeks of ARNI treatment, this patient’s plasma proBNP had decreased, while his mature BNP had increased 1.3 fold. Total plasma BNP levels were little changed. After 5 weeks of ARNI treatment, total plasma BNP, proBNP, and mature BNP had decreased with significant improvement in LVEF and a decrease in left ventricular end-diastolic volume (Figure 6).

Case 2 was an 80-year-old female patient with chronic heart failure due to an old myocardial infarction. She has a long-term history of diabetes and hypertension. She had an acute myocardial infarction (posterior infarction) several years ago and has been diagnosed with a three-vessel lesion by cardiac catheterization and underwent PCI. She has also had mild dementia for several years. Because her LVEF was 30% and plasma BNP was persistently elevated, ARNI (50 mg bid) was administered, and total BNP, proBNP, and mature BNP levels in plasma were measured before and 2, 3, and 5 weeks after initiation of ARNI treatment. After 2 weeks of ARNI treatment, proBNP was unchanged, whereas total BNP and mature BNP had increased1.5 fold and 2.3 fold, respectively. After 3 weeks of treatment, plasma total BNP, proBNP, and mature BNP had begun to decline, and they had declined further by 5 weeks, with significant improvement of LVEF and a decrease in left ventricular end-diastolic volume (Figure 6).

Case 3 was a 78-year-old male patient with chronic heart failure due to hypertensive heart disease. He has a long-term history of severe hypertension and atrial fibrillation. He has also had a history of cerebral infarction. Because his LVEF was 25% and plasma BNP was persistently elevated, ARNI (100 mg bid) was administered, and plasma total BNP, proBNP, and mature BNP were measured before and 2, 4, and 6 weeks after initiation of ARNI treatment. After 2 weeks of ARNI treatment, proBNP was little changed, whereas plasma total BNP and mature BNP were increased 1.3 and 1.5 fold, respectively. After 4 weeks of treatment, plasma total BNP, proBNP, and mature BNP had decreased, and the lower values were maintained at 6 weeks, with mild improvement of the patient’s symptoms (Figure 6).

These cases show that during ARNI treatment, plasma levels of mature BNP change over time: initially, mature BNP increases because ARNI inhibits its degradation. Thereafter, the pharmacological effect of ARNI on the heart leads to reduction of myocardial wall stress, which in turn results in reduced expression of BNP, leading to decreases in plasma mature BNP, proBNP, and total BNP. These findings suggest that decreasing myocardial wall stress with ARNI weakens the stimulus for BNP production to an extent that more than offsets the effect of inhibiting BNP degradation. Regarding the limitation of the study, in all three cases presented, the patients are thought to be responders to ARNI. On the other hand, in cases that do not respond to ARNI, persistent elevations in proBNP, mature BNP, and total BNP are expected. The extent to which the increase in plasma BNP levels seen during the early stages of ARNI administration can be regarded as an increase in mature BNP is a subject for further investigation.

## 10. Conclusions

Within the natriuretic peptide family, neprilysin has high substrate specificity for ANP and CNP but relatively weak specificity for BNP. This means that the contribution of neprilysin to the metabolic degradation of BNP is smaller than for ANP. Therefore, activation of the natriuretic peptide/cGMP pathway, which is considered to be one of the mechanisms underlying the beneficial action of ARNI, may be mediated mainly by an increase in ANP. The increase in immunoreactive BNP (total BNP) after ARNI treatment is due to an increase in mature BNP. During the initial phase of ARNI treatment, plasma ANP levels are markedly increased, while immunoreactive BNP is modestly increased due to an increase in mature BNP that reflects decreased degradation by neprilysin. Thereafter, progressive reduction of myocardial wall stress leads to decreased BNP gene expression, decreased production of BNP, and, finally, decreased plasma BNP levels. A correct interpretation of the plasma BNP levels after ARNI administration requires a comprehensive understanding of the regulatory mechanisms affecting BNP gene expression; protein production, secretion, and metabolism; the molecular forms of BNP; the immunoassays used to measure BNP; and the substrate specificity of neprilysin for BNP.

In conclusion, BNP is considered a useful biomarker for evaluating heart failure with or without ARNI and should continue to be used for this purpose in the “ARNI era”.

## Figures and Tables

**Figure 1 biology-11-01034-f001:**
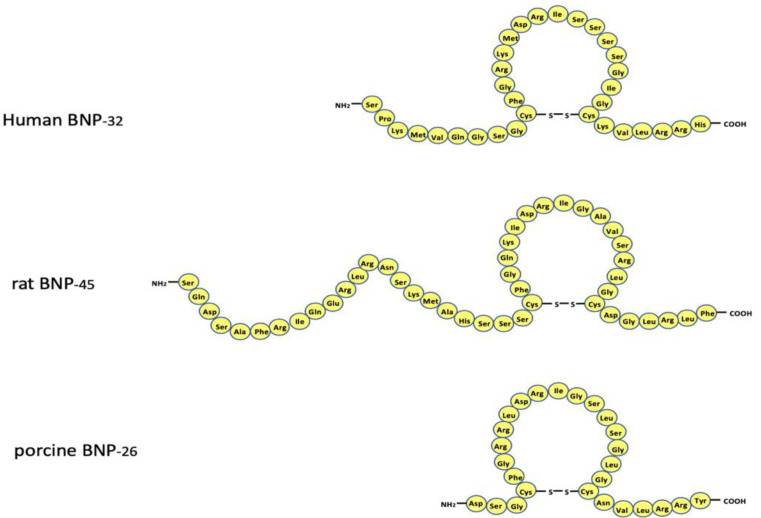
Amino acid sequence of circulating human, rat, and porcine B-type natriuretic peptide. Note that the amino acid sequence varies greatly among the species. The reason is described in the text.

**Figure 2 biology-11-01034-f002:**
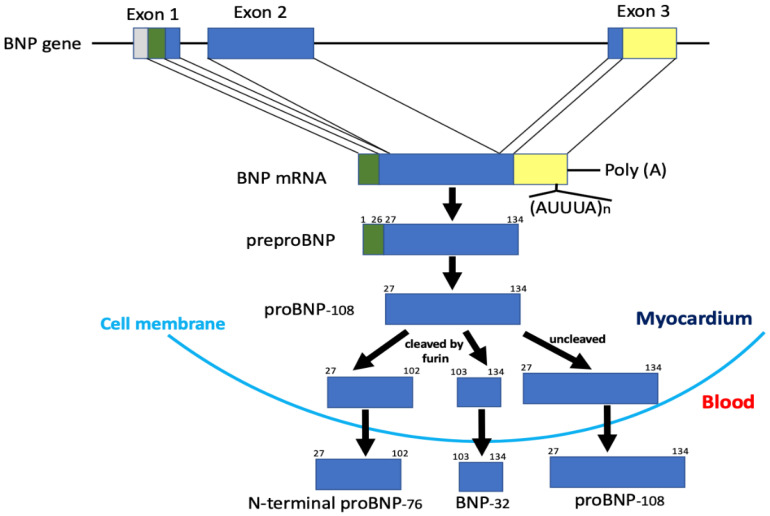
Structure of the gene and biosynthetic pathway of human BNP. Note that only a portion of proBNP is cleaved in the myocardium and then proBNP, mature BNP, and N-terminal proBNP are all secreted into the blood.

**Figure 3 biology-11-01034-f003:**
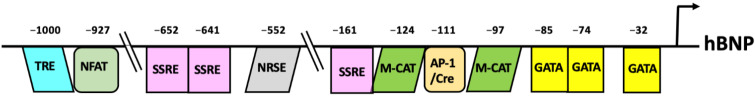
Schematic representation of the human BNP promoters showing the cis-acting elements identified so far.

**Figure 4 biology-11-01034-f004:**
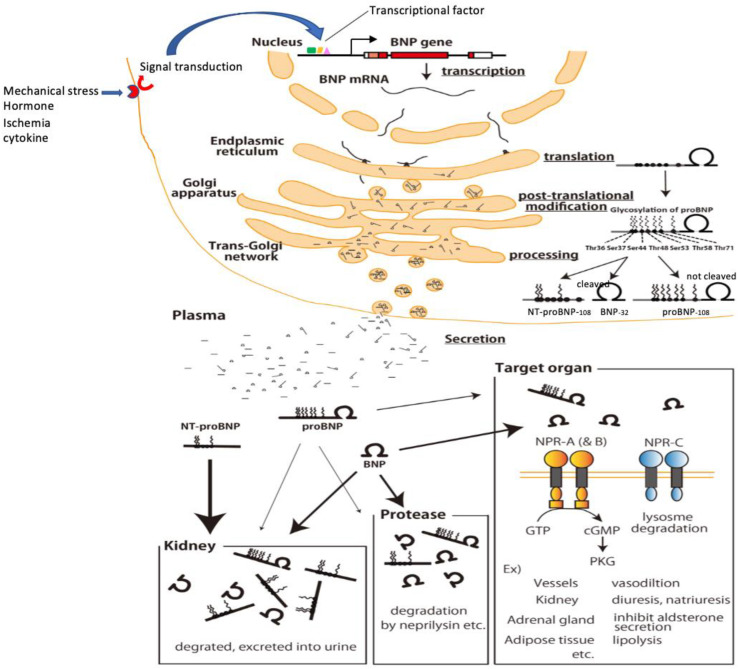
Production, processing, secretion and metabolism of BNP in cardiomyocytes. Mechanical stress, ischemia, hormones, cytokines, etc. stimulate the receptors and enhance BNP gene expression via signal transduction/transcription factors. The enhanced BNP mRNA, which undergoes splicing, crosses the nuclear membrane and is translated in the endoplasmic reticulum to produce preproBNP. The signal peptide is then removed by signalpeptidase, yielding proBNP. As proBNP is transported through the Golgi apparatus, glycosylation occurs at seven sites in the N-terminal region, and within the trans-Golgi network some of the proBNP is cleaved by furin into BNP and NT-proBNP (a portion remains as proBNP). Then, BNP, NT-proBNP, and proBNP are released into the blood. Circulating BNP binds to its receptors on various tissues and organs throughout the body (blood vessels, kidneys, adrenal glands, fat cells, etc.), after which it is internalized by the cell and metabolized. Circulating BNP is also degraded in the blood by proteases (neprilysin and others), and it is also filtered by the glomerulus, metabolized, and excreted in the urine. Binding of circulating proBNP to its receptor is weak, is not degraded by neprilysin and is less filtrated from the glomerulus. On the other hand, because circulating NT-proBNP does not bind to BNP receptors, nor is it degraded by enzymes such as neprilysin, its clearance depends almost exclusively on excretion via the kidney. NT-proBNP levels therefore increase sharply with even mild renal dysfunction and are markedly increased in renal failure.

**Figure 5 biology-11-01034-f005:**
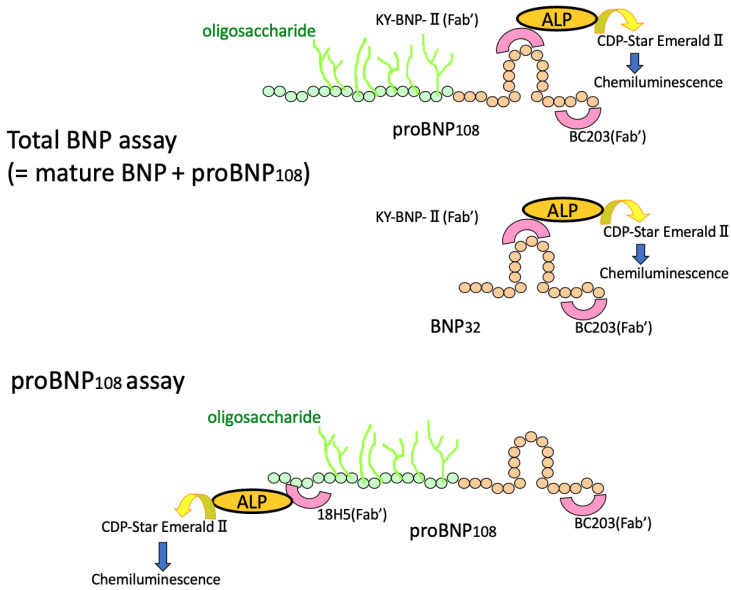
Schematic diagram of the total BNP and proBNP assay systems. BC203, a monoclonal antibody that specifically recognizes the C-terminus of BNP, is used as a capture antibody for both total BNP and proBNP. KY-BNP-II (Fab’), a monoclonal antibody that specifically recognizes the ring structure of BNP, is used as a detection antibody for total BNP assay. 18H5 (Fab’), a monoclonal antibody that specifically recognizes the N-terminus of proBNP, is used as a detection antibody for proBNP assay. ALP: Alkaline phosphatase; CDP-Star EmeraldII (Chemiluminescent Substrate): Disodium2-chloro-5-(4-methoxy-spiro{1,2-dioxetane-3,2′-(5′-chloro)-tricyclo[3,3,1,13,7]decan}-4-yl)-1-phenyl phosphate.

**Figure 6 biology-11-01034-f006:**
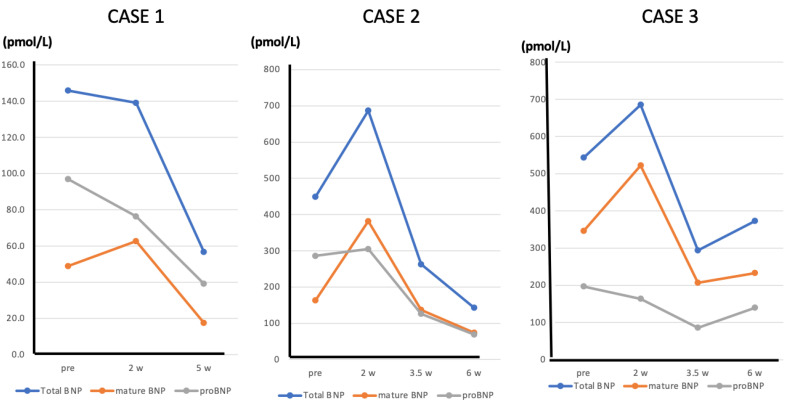
Line graphs showing the changes in total BNP, mature BNP, and proBNP in Cases 1, 2, and 3 before and after initiating ARNI administration.

**Table 1 biology-11-01034-t001:** Types of antibodies and amino acid recognition sites for various BNP assay kits.

Immunoassay/Instrument	Capture Antibody	Detection Antibody	Standard
Abbott Architect BNP	Mouse monoclonal antibody aa5–13	Mouse monoclonal antibody aa26–32	Synthetic BNP-32
Alere/Quidel Triage BNP	Mouse monoclonal antibody aa5–13	Omniclonal antibody (epitope not characterized)	Recombinant BNP-32
Siemens Centaur BNP	Mouse monoclonal antibody aa27–32	Mouse monoclonal antibody aa14–21	Synthetic BNP-32
Pylon Single Epitope BNP	Mouse monoclonal antibody aa11–17	Recognizes the immune complex of capture antibody with BNP/proBNP	Glycosylated proBNP

## Data Availability

Not applicable.

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
