# Peer review of "B-Type Natriuretic Peptide (BNP) Revisited—Is BNP Still a Biomarker for Heart Failure in the Angiotensin Receptor/Neprilysin Inhibitor Era?"

_biology, 2022, doi:10.3390/biology11071034_

Round 1
Reviewer 1 Report
The authors make a “state of the art” review on the knowledge regarding BNP and its interaction with ACEI and ARNI. BNP is a useful biomarker for evaluating heart failure, and a description of the physiological pathway which might interfere with its values and tests, with or without ARNI.
Although extensive and very detailed, the manuscript does not present the methodology corresponding to a systematic review. Corresponds to a narrative review of BNP phisiology
The authors fail to quantify and summarize the direct clinical implications of the ARNI on BNP values.
Minor reviews.
Line 111
“proATP is cleaved into ANP and NT-proANP”
Please review proATP to proANP
The letter type and spacing is not consistent during the manuscript.
Line 259
“On the other hand, the same treatment does not affect BMP-induced cGMP formation”
Please review BMP to BNP
Author Response
To the reviewer 1
Comments and Suggestions for Authors
The authors make a “state of the art” review on the knowledge regarding BNP and its interaction with ACEI and ARNI. BNP is a useful biomarker for evaluating heart failure, and a description of the physiological pathway which might interfere with its values and tests, with or without ARNI.
- Although extensive and very detailed, the manuscript does not present the methodology corresponding to a systematic review. Corresponds to a narrative review of BNP phisiology
Thank you for your insightful review and comments. This manuscript is submitted as ‘review’ section, not as ‘systematic review’ section. Therefore, this is just a narrative review and does not include methodology.
- The authors fail to quantify and summarize the direct clinical implications of the ARNI on BNP values.
Thank you for your comment. This manuscript is not ‘systematic review’ and this is just a narrative review. The published results are currently limited to quantify the direct clinical implications of the ARNI on BNP values. Our object of this manuscript is to provide the concept of clinical implications of the ARNI on BNP values to the readers based on the knowledge of biochemistry, molecular biology, pharmacology, physiology and laboratory medicine.
Minor reviews.
Line 111
“proATP is cleaved into ANP and NT-proANP” Please review proATP to proANP
Thank you very much. We corrected it as you suggested (proANP)(line 119).
The letter type and spacing is not consistent during the manuscript.
Thank you for your comments. We carefully checked the letter type and spacing throughout the manuscript.
Line 259
“On the other hand, the same treatment does not affect BMP-induced cGMP formation”
Please review BMP to BNP
We corrected it as you suggested (line 285). Thank you.
Reviewer 2 Report
An interesting and informative review of the physiology and pathology of the natriuretic peptide system, neprilysin, and ARNI effects. The article can be published in the presented version
Author Response
An interesting and informative review of the physiology and pathology of the natriuretic peptide system, neprilysin, and ARNI effects. The article can be published in the presented version
Thank you very much for your good comment to our review.
Reviewer 3 Report
Dear Authors, I have read your manuscript with interest.
The current manuscript titled: "B-type natriuretic peptide (BNP) revisited – Is BNP still a biomarker for heart failure in the angiotensin receptor/neprilysin inhibitor era?" represents an important analysis of evolving field of Cardiology and Laboratory Medicine.
The title reflects the manuscript content and helps the reader navigate the article essence.
The abstract contains all the necessary information in a concise form.
In my opinion, these is the adjustment which should be made to increase the value of your manuscript:
1. In the manuscript title, please change all the letters to capitals.
2. Line 21: add please abbreviation for BNP.
3. Line 25: add please abbreviation for NPR.
4. Line 27: add please abbreviation for ANP.
5. Line 30: add please abbreviation for ARNI.
6. In the Introduction section, before focusing on the importance of BNP/NT-proBNP in heart failure, please describe all the normal and pathological patient conditions in which BNP/NT-proBNP may be elevate.
7. Also, it would be very useful to add information about age differences in BNP/NT-proBNP values and the practical significance of using these BNP/NT-proBNP values ranges.
8. In the general manuscript text, as well as in the figures and tables, a different font is used, please homogenize the font.
9. In clinical cases, please add more information about the comorbidities of the presented patients.
10. For further improving the manuscript content, I suggest the Authors to read and consider relevant manuscript in this field: doi: 10.1177/0300060518798257.
11. Line 462: move 10. Conclusions one line down.
12. The manuscript contains some punctuation errors, please revise the text.
Author Response
To the reviewer 3
Dear Authors, I have read your manuscript with interest.
The current manuscript titled: "B-type natriuretic peptide (BNP) revisited – Is BNP still a biomarker for heart failure in the angiotensin receptor/neprilysin inhibitor era?" represents an important analysis of evolving field of Cardiology and Laboratory Medicine.
The title reflects the manuscript content and helps the reader navigate the article essence.
The abstract contains all the necessary information in a concise form.
In my opinion, these is the adjustment which should be made to increase the value of your manuscript:
- In the manuscript title, please change all the letters to capitals.
According to your suggestions, we change all the letters to capitals in title (Line 2-4).
- Line 21: add please abbreviation for BNP.
According to your suggestions, we added the abbreviation for B-type (or brain) natriuretic peptide (BNP) gene (Line 20-21).
- Line 25: add please abbreviation for NPR.
According to your suggestions, we added the abbreviation for natriuretic peptide receptor (NPR) (Line 24-25).
- Line 27: add please abbreviation for ANP.
According to your suggestions, we added the abbreviation for atrial natriuretic peptide (ANP) (Line 27).
- Line 30: add please abbreviation for ARNI.
According to your suggestions, we added the abbreviation for angiotensin receptor neprilysin inhibitor (ARNI) (Line 28).
- In the Introduction section, before focusing on the importance of BNP/NT-proBNP in heart failure, please describe all the normal and pathological patient conditions in which BNP/NT-proBNP may be elevate.
According to your suggestion, we rewrote the introduction section. We added new sentences (line 43-46) and 49-52) and new references (Ref.3-7 and Ref.12-13).
- Also, it would be very useful to add information about age differences in BNP/NT-proBNP values and the practical significance of using these BNP/NT-proBNP values ranges.
According to your suggestion, we added new sentences (line 52-59) and new references (14-18).
- In the general manuscript text, as well as in the figures and tables, a different font is used, please homogenize the font.
According to your suggestion, we unified font in all the figures and table.
- In clinical cases, please add more information about the comorbidities of the presented patients.
According to your suggestion, we added more information about the comorbidities of the presented three cases (Case 1: line 445-449; Case 2: line 458-461; Case 3: 469-470).
- For further improving the manuscript content, I suggest the Authors to read and consider relevant manuscript in this field: doi: 10.1177/0300060518798257.
We read the paper with great interest. Since the paper you suggested has a slightly different research purpose from the current review, we will cite it in our next study on the diagnosis of heart failure using BNP molecular forms.
- Line 462: move 10. Conclusions one line down.
Thank you. We corrected as you suggested (line 491).
- The manuscript contains some punctuation errors, please revise the text.
We revised some punctuation errors throughout the manuscript.
Reviewer 4 Report
The aurhors assessed the significance of BNP as a marker for heart failure in the ARNI era. They concluded that BNP is considered to be a useful biomarker for evaluating heart failure with or without ARNI and should continue to be used for this purpose in the “ARNI era.”
I have the following concerns:
1. Please include the Limitation section.
2. Could you please comment on practical implications of the study?
3. Please create a Table with clinical studies on BNP including type of the study, number of patients, etc.
Author Response
Reviewer 4
The authors assessed the significance of BNP as a marker for heart failure in the ARNI era. They concluded that BNP is considered to be a useful biomarker for evaluating heart failure with or without ARNI and should continue to be used for this purpose in the “ARNI era.”
I have the following concerns:
1. Please include the Limitation section.
Since this manuscript is a review article, we do not usually have a limitation section. Section 9, however, is not a section of the review. We have shown changes in the molecular forms of BNP in three cases who responded to ARNI. Since we could not describe the changes in BNP molecular forms in the cases who did not respond to ARNI, so we have included this as a limitation of the study (line 483-487).
- Could you please comment on practical implications of the study?
Although ARNI is effective in mass studies when administered to patients with heart failure, in individual cases, it can be difficult to determine its efficacy. If plasma BNP is measured during ARNI administration to determine its efficacy, the understanding of the mRNA expression regulation, metabolism, and assay system of BNP described in this review make readers easily judgement of its efficacy using BNP after ARNI treatment, contributing to better practice of clinical cardiology.
- Please create a Table with clinical studies on BNP including type of the study, number of patients, etc.
There are over 200 published papers on Saccubitril/valsartan for BNP, and it is not the intent of this REVIEW to represent them all in a table. The purpose of this review is to provide the readers with the mechanisms by which plasma BNP is decreased or increased, and how plasma BNP may be altered by ARNI administration, from the perspective of biochemistry, physiology, molecular biology, laboratory medicine, etc. The table that the reviewer wishes to see may be published in a meta-analysis or other paper in the near future.
Round 2
Reviewer 3 Report
The changes made have significantly increased the manuscript quality.
I recommend this article for publication.
Good luck!